# Structural basis of G protein–Coupled receptor CMKLR1 activation and signaling induced by a chemerin-derived agonist

Xuan Zhang[1☉], Tina Weiß[2☉], Mary Hongying Cheng[3,4☉], Siqi Chen[5☉], Carla Katharina Ambrosius[2], Anne Sophie Czerniak[2], Kunpeng Li[6], Mingye Feng[5]*, Ivet Bahar[3,4,7]*, Annette G. Beck-Sickinger[2]*, Cheng Zhang[1]*

**1** Department of Pharmacology and Chemical Biology, School of Medicine, University of Pittsburgh, Pittsburgh, Pennsylvania, United States of America, **2** Institute of Biochemistry, Faculty of Life Sciences, Leipzig University, Leipzig, Germany, **3** Department of Computational and System Biology, School of Medicine, University of Pittsburgh, Pittsburgh, Pennsylvania, United States of America, **4** Laufer Center for Physical and Quantitative Biology, Stony Brook University, Stony Brook, New York, United States of America, **5** Department of Immuno-Oncology, Beckman Research Institute, City of Hope Comprehensive Cancer Center, Duarte, California, United States of America, **6** Cryo-EM core facility, Case Western Reserve University, Cleveland, Ohio, United States of America, **7** Department of Biochemistry and Cell Biology, School of Medicine, Stony Brook University, Stony Brook, New York, United States of America

☉ These authors contributed equally to this work.
* mfeng@coh.org (MF); bahar@laufercenter.org (IB); abeck-sickinger@uni-leipzig.de (AGB-S); chengzh@pitt.edu (CZ)

**Data Availability Statement:** The 3D cryo-EM density map of CMKLR1-Gi-scFv16 complex with chemerin9 has been deposited in the Electron Microscopy Data Bank under the accession

## Abstract

Chemokine-like receptor 1 (CMKLR1), also known as chemerin receptor 23 (ChemR23) or chemerin receptor 1, is a chemoattractant G protein–coupled receptor (GPCR) that responds to the adipokine chemerin and is highly expressed in innate immune cells, including macrophages and neutrophils. The signaling pathways of CMKLR1 can lead to both pro- and anti-inflammatory effects depending on the ligands and physiological contexts. To understand the molecular mechanisms of CMKLR1 signaling, we determined a high-resolution cryo-electron microscopy (cryo-EM) structure of the CMKLR1-G$_i$ signaling complex with chemerin9, a nanopeptide agonist derived from chemerin, which induced complex phenotypic changes of macrophages in our assays. The cryo-EM structure, together with molecular dynamics simulations and mutagenesis studies, revealed the molecular basis of CMKLR1 signaling by elucidating the interactions at the ligand-binding pocket and the agonist-induced conformational changes. Our results are expected to facilitate the development of small molecule CMKLR1 agonists that mimic the action of chemerin9 to promote the resolution of inflammation.

## Introduction

Chemerin is a 163-amino acid preprotein encoded by the *retinoic acid receptor responder protein 2* (*RARRES 2*) gene that is up-regulated by the retinoid drug tazarotene [1–4]. Cleavage of the N-terminal 20-amino acid signal peptide and the C-terminal 6-amino acid segment leads

numbers EMD-40450. Atomic coordinates for the atomic model have been deposited in the Protein Data Bank (PDB) under the accession numbers 8SG1. The FACS data has been deposited to FlowRepository with the ID FR-FCM-Z6US (http://flowrepository.org/id/FR-FCM-Z6US).

**Funding:** Grants from the National Institutes of Health (NIH) in the US R35GM128641 (to C.Z.), R01CA255250 (to M.F.), R01CA258778 (to M.F.), and R01GM139297 (to I.B.). The German Research Foundation (Deutsche Forschungsgemeinschaft, DFG) project number 209933838, CRC1052/3, C08 (to A.G.B-S.). The funders had no role in study design, data collection and analysis, decision to publish, or preparation of the manuscript.

**Competing interests:** The authors declare no competing financial interests.

**Abbreviations:** APC, antigen-presenting cell; ChemR23, chemerin receptor 23; CHS, cholesteryl hemisuccinate; CMKLR1, chemokine-like receptor 1; cryo-EM, cryo-electron microscopy; CTF, contrast transfer function; C5aR, C5a receptor; DC, dendritic cell; ECL2, extracellular loop 2; eYFP, enhanced yellow fluorescent protein; FPR2, formyl peptide receptor 2; FSC, Fourier shell correlation; GPCR, G protein–coupled receptor; ICL2, intracellular loop2; ICL3, intracellular loop 3; LMNG, lauryl maltose neopentylglycol; MD, molecular dynamics; MM/GBSA, Molecular Mechanics/Generalized Born Surface Area; MR, mannose receptor; PAM, positive allosteric modulator; PUFA, polyunsaturated fatty acid; RARRES 2, retinoic acid receptor responder protein 2; RMSD, root-mean-square deviation; RvE1, resolvin E1; SPM, specialized pro-resolvin lipid mediator; SPPS, solid phase peptide synthesis; wt, wild-type; 7-TMs, 7 transmembrane helices.

to the most active form of chemerin containing residues 21–157, which functions as the endogenous ligand to activate the chemokine-like receptor 1 (CMKLR1) or chemerin receptor 23 (ChemR23) or chemerin receptor 1 [2–5]. Other isoforms of chemerin due to cleavage at different sites of the C-terminal region have also been detected in vivo with lower potencies in activating CMKLR1 compared to the 21–157 isoform [1]. CMKLR1 belongs to a family of $G_i$-coupled chemoattractant G protein–coupled receptors (GPCRs) in the γ-subgroup of the Class A GPCRs [6]. Other members of this family as close phylogenetic neighbors of CMKLR1 include receptors for anaphylatoxin C5a, formyl peptides, and prostaglandin D2 ($PGD_2$) [6].

Chemerin is usually considered as an adipocytokine or adipokine since it is produced by adipocytes and can act on CMKLR1 to regulate adipogenesis and energy metabolism in adipose tissue [1,4,7,8]. Increased levels of chemerin and CMKLR1 have been linked to obesity and insulin resistance [3,4,8]. In addition, the chemerin-CMKLR1 signaling axis also plays pleiotropic roles in inflammation. Chemerin can act as a potent chemoattractant to enhance chemotaxis of monocytes especially dendritic cells (DCs) through CMKLR1 signaling [4,7,9,10]. In this respect, the activation of CMKLR1 promotes the onset of inflammation. On the other hand, numerous studies suggested that the signaling of CMKLR1 could also resolve inflammation [1,7,9]. Such multifaceted functional roles in inflammation are also observed for another chemoattractant GPCR, the formyl peptide receptor 2 (FPR2) [11,12], which is closely related to CMKLR1.

The apparent contradictory functional roles of CMKLR1 in inflammation may be associated with the spatiotemporal regulation of the inflammation processes in cells. It is possible that in the early stage of inflammation, the CMKLR1 signaling is mainly pro-inflammatory by promoting chemotaxis and activation of DCs and macrophages, whereas at the late stage of inflammation, CMKLR1 activation can induce pro-resolving pathways to dampen inflammation. Another possibility for the differing effects of CMKLR1 signaling on inflammation is that different chemerin isoforms may activate CMKLR1 to induce distinct signaling outcomes [1]. Indeed, 2 synthetic peptides, chemerin9 [13,14] and chemerin15 [15] corresponding to the 149–157 and the 141–155 amino acid segments of chemerin, respectively, have been characterized as stable agonists of CMKLR1 that induce anti-inflammatory effects [9].

Chemerin9 has been tested and shown positive therapeutic effects in several animal disease models of cardiovascular diseases [16,17], memory impairment [18], and diabetes [19]. Interestingly, recent studies provided strong evidence indicating that the activation of CMKLR1 mediates the protective effects of ω3-polyunsaturated fatty acids (PUFAs) in aortic valve stenosis [18], atherosclerosis [19], pulmonary hypertension [20], and depression [21,22], which involve resolvin E1 (RvE1) [23,24]. RvE1 is a specialized pro-resolvin lipid mediator (SPM) that has been suggested to act on CMKLR1 to promote the resolution of inflammation [25–27]. However, it is not clear whether RvE1 can directly bind to and activate CMKLR1 to induce $G_i$ signaling.

The potential roles of CMKLR1 signaling in inflammation may provide promising new opportunities for developing anti-inflammatory drugs. However, although there are plenty of peptide agonists derived from chemerin, the limited availability of synthetic ligands of CMKLR1 has impeded pharmacological investigation and drug development in this area. One potent antagonist of CMKLR1 named CCX832 has been identified, but the chemical structure remains undisclosed. Additionally, a few 2-aminobenzoxazole analogues have been reported as CMKLR1 inhibitors [28,29]. Nevertheless, the only commercially available small-molecule CMKLR1 antagonist is α-NETA, which can target other proteins and shows low micromolar potency against CMKLR1 [30]. As to CMKLR1 agonists, several chemerin-derived peptide agonists have been developed [1,7,9], and one CMKLR1 antibody that functions as a receptor agonist was reported to promote inflammation resolution in chronic colitis models [31]. Yet,

no small-molecule agonists of CMKLR1 have been reported so far. In contrast, numerous small-molecule agonists have been developed for the closely related pro-resolving GPCR, FPR2, as novel anti-inflammatory agents for clinical investigation [12].

To address those questions in CMKLR1 physiology, pharmacology, and drug development, we mainly focused on the peptide agonists and solved a cryo-electron microscopy (cryo-EM) structure of the CMKLR1-$G_i$ signaling complex with chemerin9, which, in our assays, induced a phenotype of macrophages that does not fall into the oversimplified M1- nor M2-macrophages paradigm [32]. Together with molecular dynamics (MD) simulations and mutagenesis studies, our structure revealed critical molecular features for the binding of chemerin9 and shed light on the molecular mechanism by which chemerin9 activates CMKLR1 to promote $G_i$ signaling. The structural information is anticipated to facilitate the development of synthetic small molecule agonists mimicking the action of chemerin9 for CMKLR1.

## Results

### Complex phenotypic changes of macrophages induced by chemerin9

Despite the positive therapeutic effects of chemein9 in multiple disease models, it remains unclear whether chemerin9 can induce an anti-inflammatory phenotype of macrophages. To investigate the functional effects of the chemerin9-CMKLR1 signal axis in macrophages, we treated primary human macrophages with chemerin9, along with IFNg and LPS as the reference signal molecules to induce the M1-like macrophages and IL-10 as the reference signal molecule to induce the M2-like macrophages. We then checked the expression levels of 4 cell surface proteins, HLA-DR, CD206, CD163, and CD86 (**Figs 1A and S1**), which are characteristic markers of different phenotypes of macrophages [33,34]. HLA-DR is a class II major histocompatibility complex cell surface molecule, a signature cell marker for antigen-presenting cells (APCs), while CD86 is a selective ligand for the costimulatory molecule CD28, the activation of which is indispensable for optimal T cell prime and activation; together, HLA-DR and CD86 are signature cell surface markers for the canonical M1-like macrophages [34]. On the other hand, CD206, the mannose receptor (MR), is a glycosylated surface protein belonging to the scavenger receptor family. Similarly, CD163, another member of the scavenger receptor family, is the receptor for hemoglobin-haptoglobin complex. The up-regulation of

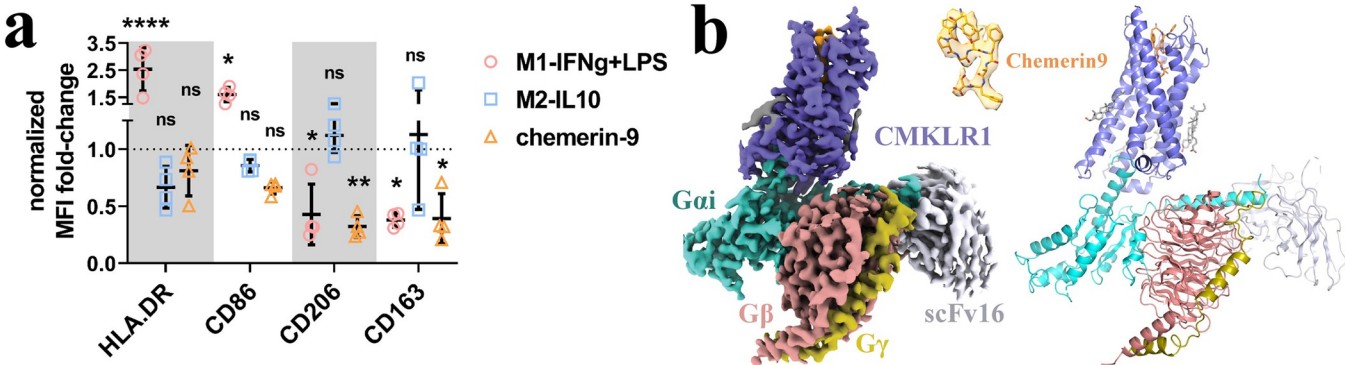

**Fig 1. Chemerin9-induced phenotypic changes of macrophage and overall structure of the CMKLR1-$G_i$-scFv16 complex with chemerin9. (a)** The MFI fold-change of HLA-DR, CD86, CD206, and CD163 in primary macrophages under various stimulation conditions, normalized to the M0 condition (medium). Two-way ANOVA with Dunnett's multiple comparisons test, with M0 group as control reference. $n = 4$, each dot represents macrophages from an independent donor. ns, not significant, *, $p < 0.05$, **, $p < 0.01$, ****, $p < 0.0001$ The underlying data for Fig 1A can be found in S1 Data. **(b)** The left and right panels show the cryo-EM density map and the overall structure, respectively. CMKLR1 is colored in blue. Chemerin9 is colored in orange. Gαi, Gβ, and Gγ subunits are colored in cyan, salmon, and dark yellow, respectively. ScFv16 is colored in grey. CMKLR1, chemokine-like receptor 1; cryo-EM, cryo-electron microscopy; MFI, mean fluorescence intensity.

CD206 and CD163 in macrophages has been associated with the anti-inflammatory and tissue-repair phenotype [33,35].

In our assays, IFNg plus LPS treatment led to elevated expression of HLA-DR and CD86 and lowered the expression of CD206 and CD163 in macrophages (**Fig 1A**), suggesting the pro-inflammatory M1-like phenotype [32]. IL-10 treatment induced the down-regulation of HLA-DR and CD86 and the up-regulation of CD206 and CD163 in primary human macrophages (**Fig 1A**), consistent with an anti-inflammatory phenotype, although the changes were statistically insignificant. When stimulated by chemerin9, the primary human macrophages showed significantly decreased levels of CD206 and CD163 as done by IFNg plus LPS, which induced inflammation (**Fig 1A**). However, chemerin9 did not induce the up-regulation of HLA-DR or CD86, which is a signature feature of the pro-inflammatory macrophages. In fact, we observed slightly lowered levels of these 2 markers upon chemerin9 stimulation compared to those induced by IL-10, consistent with an anti-inflammatory effect (**Fig 1A**). Taken together, the results suggest that the chemerin9-CMKLR1 signaling axis induced a phenotype of macrophages that does not simply fall into the oversimplified classification of the M1- and the M2-macrophages paradigm. How CMKLR1 regulates inflammation via action on macrophages needs further investigation.

## Structure determination and overall structure of the chemerin9-CMKLR1-G$_i$ complex

To assemble the CMKLR1 and G$_i$ complex, we used the wild-type (wt) human CMKLR1 and human G$_i$ heterotrimer containing the G$_{\alpha i}$, G$_{\beta 1}$, and G$_{\gamma 2}$ subunits, similarly to what we published previously in determining the structure of G$_i$-coupled FPRs [36,37]. To facilitate the expression of CMKLR1 in insect cells, we fused a peptide derived from the N-terminal 28-amino acid segment of human β2-adrenergic receptor to the N-terminal end of CMKLR1. An antibody fragment, scFv16, was used to stabilize the G$_i$ heterotrimer [38]. The structure of the CMKLR1-G$_i$-scFv16 complex with chemerin9 was determined by cryo-EM to an overall resolution of 2.94 Å (**Figs 1B** and **S2 and Table 1**). Most regions of CMKLR1 (from P33 to S332 including the 7 transmembrane helices (7-TMs) and the intracellular loops 2 and 3 (ICL2 and ICL3)) could be modeled except for a part of the extracellular loop 2 (ECL2) due to weak cryo-EM density. The clear density of the peptide ligand allowed unambiguous modeling of all 9 amino acids from Tyr149 to Ser157 of chemerin9 (residues in chemerin9 are referred to by 3-letter names, and residues in CMKLR1, G protein, and other GPCRs are referred to by 1-letter names hereafter) (**Fig 1B**).

CMKLR1 is closely related to the C5a receptor (C5aR) and FPRs, all of which are G$_i$-coupled chemoattractant GPCRs with peptide or protein endogenous ligands [6]. The active structure of CMKLR1 is highly similar to that of C5aR [39], FPR1, and FPR2 [36,37,40], as indicated by the root-mean-square deviation (RMSD) in the Cα atoms at 1.219 Å, 1.284 Å, and 1.048 Å, respectively (**Fig 2A**). The cytoplasmic regions of these 3 receptors, including ICL1-3, can be well aligned (**Fig 2B**), indicating a highly conserved mechanism of receptor activation and G$_i$-coupling. However, the extracellular regions of these chemotactic receptors exhibit significant structural differences (**Fig 2B**). Accordingly, the binding pose of chemerin9 is distinct from that of the peptide agonist fMLFII in FPR1 and FPR2, as well as that of the C-terminal region of C5a in C5aR [39,41] (**Fig 2C**).

In our structure, we also observed strong cryo-EM densities on the surface of the 7-TMs likely for lipids (**S3A Fig**). To fit the densities, we modeled palmitic acid and cholesterol molecules (**S3A Fig**). Notably, the 2 palmitic acid molecules are located in a groove above ICL2 among TM3, 4, and 5 (**S3B Fig**). This pocket shares high similarities with the allosteric sites in

**Table 1. cryo-EM data collection and refinement statistics.**

| | Chemerin9-CMKLR1-$G_i$ complex (EMD-40450, PDB 8SG1) |
|---|---|
| Data collection and processing | |
| Magnification | 105,000 |
| Voltage (kV) | 300 |
| Electron exposure ($e^-$/Å$^2$) | 56 |
| Defocus range (μm) | −1.0 to −1.8 |
| Pixel size (Å) | 0.848 |
| Symmetry imposed | C1 |
| Initial particle images (no.) | 3,900,728 |
| Final particle images (no.) | 242,745 |
| Map resolution (Å) | 2.94 |
| FSC threshold | 0.143 |
| Map resolution range (Å) | 2.0–4.0 |
| Refinement | |
| Model resolution (Å) | 3.2 |
| FSC threshold | 0.5 |
| Model composition | |
| Non-hydrogen atoms | 8,846 |
| Protein residues | 1,127 |
| Ligand | 1 |
| Lipids | 3 |
| R.m.s. deviations | |
| Bond lengths (Å) | 0.005 |
| Bond angles (°) | 0.888 |
| Validation | |
| MolProbity score | 1.60 |
| Clashscore | 6.41 |
| Rotamer outliers (%) | 1.78 |
| Ramachandran plot | |
| Favored (%) | 97.74 |
| Allowed (%) | 2.26 |
| Disallowed (%) | 0 |

C5aR [41] and GPR40 [42] (**S3B Fig**). In particular, one of the palmitic acid molecules aligns well with the GPR40 positive allosteric modulator (PAM) AP8 when the structures of the 2 receptors are superimposed (**S3B Fig**). This suggests the possibility of developing PAMs for CMKLR1 that target this pocket, potentially offering a new therapeutic avenue for this receptor.

## An "S-shape" binding mode of chemerin9 to induce CMKLR1 activation

Chemerin9 adopts an "S-shape" binding mode to occupy a binding pocket in CMKLR1 that is highly open to the extracellular milieu (**Fig 2C and 2D**). The side chain of the first residue of chemerin9, Tyr149, folds back towards the binding pocket. Phe150 of chemerin9 form π–π interaction with F190 from the ECL2 of CMKLR1 (**Fig 2E**). It is to be noted that the density of Phe150 of chemerin9 is weak, which may suggest that the π–π interaction with F190 of CMKLR1 is not strong enough to stabilize the residue in one conformation. Following a turn caused by Pro151, the C-terminal segment of chemerin9 from Gly152 to Ser157 forms a cyclic peptide-like conformation to insert into the 7TM bundle of CMKLR1 (**Fig 2E**). A polar

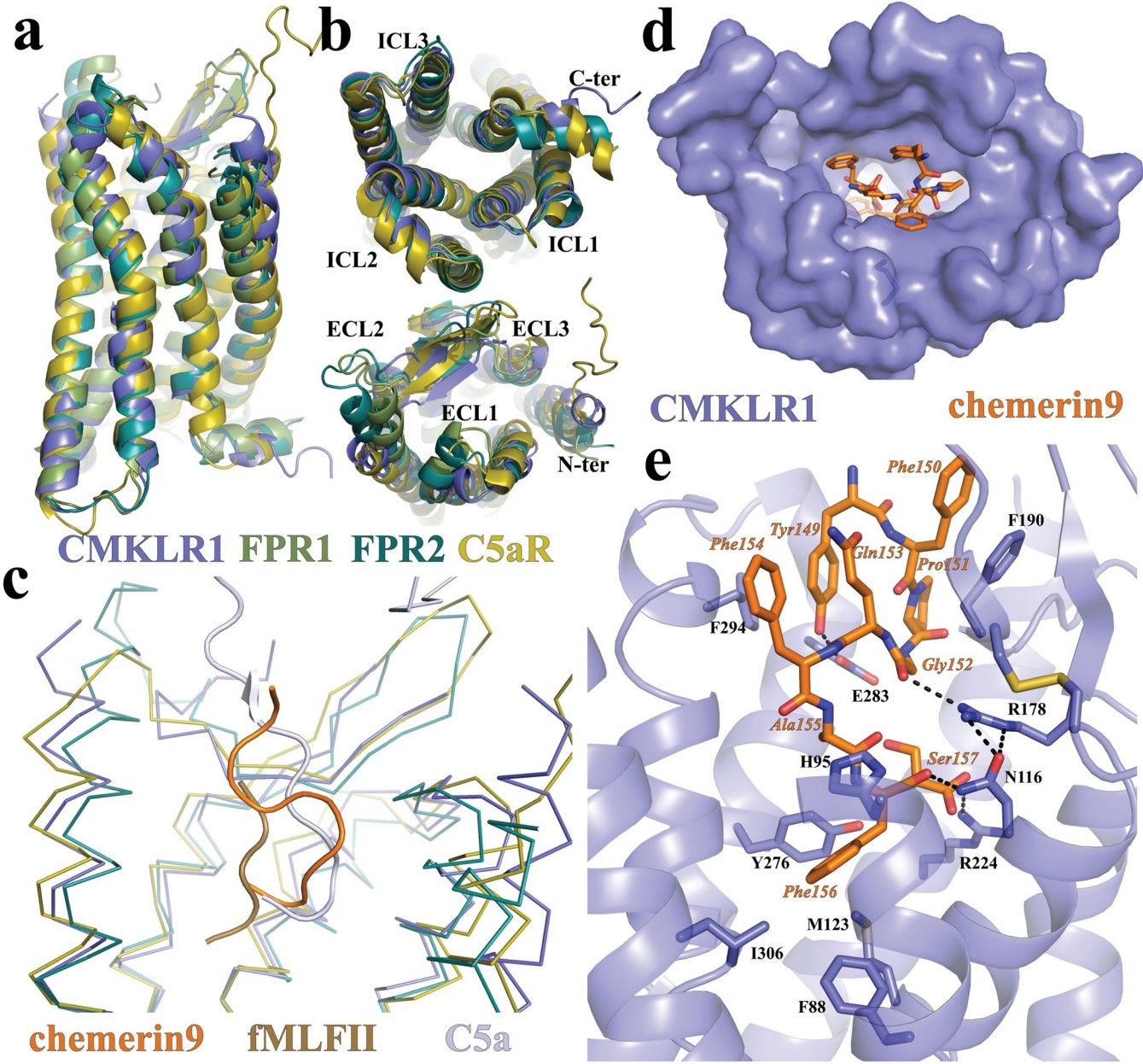

**Fig 2. Structural comparison of CMKLR1 with other chemotactic GPCRs and chemerin9 binding. (a** and **b)** Superimposition of the structures of active CMKLR1, FPR1 (PDB ID 7T6T), FPR2 (PDB ID 7T6V), and C5aR (PDB ID 7Y65). **(c)** Binding poses of chemerin9, fMLFII, and C5a as the peptide agonists of CMKLR1, FPR1, and C5aR, respectively. **(d)** Chemerin9 binding pocket viewed from the extracellular surface. **(e)** Interactions between chemerin9 and CMKLR1. The polar interactions are shown as dashed lines. CMKLR1, chemokine-like receptor 1; C5aR, C5a receptor; FPR1, formyl peptide receptor 1; FPR2, formyl peptide receptor 2; GPCR, G protein–coupled receptor.

interaction network is formed among the carbonyl groups of Phe156 and Gly152 of chemerin9 and CMKLR1 residues N116$^{3.29}$ and R178$^{4.64}$ (Ballesteros–Weinstein numbering [43]) (**Fig 2E**). The carboxylate group of Ser157 and the side chain of Tyr149 form hydrogen bonds with the side chains of R224$^{5.42}$ and E283$^{6.58}$ of CMKLR1, respectively. In addition to the polar interactions, Phe156 of chemerin9 also engages in hydrophobic and π–π interactions with surrounding CMKLR1 residues F88$^{2.53}$, H95$^{2.60}$, M123$^{3.36}$, Y276$^{6.51}$, and I306$^{7.43}$ (**Fig 2E**).

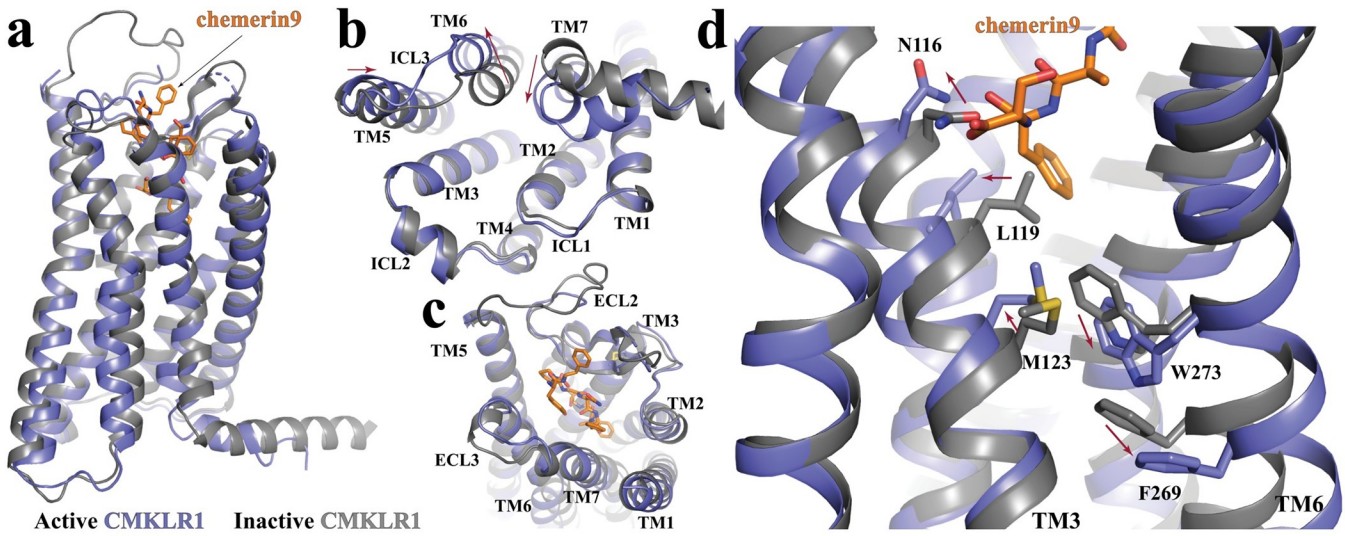

**Fig 3. CMKLR1 activation. (a)** Superimposition of the active CMKLR1 structure (blue) to the Alphafold2-predicted inactive GPR84 structure (dark grey). The intracellular and extracellular regions are shown in **(b)** and **(c)**, respectively. **(d)** Residues involved in the receptor activation at the core region of CMKLR1. The red arrows indicate conformational changes from the inactive to the active conformation.

Since there is no structure of inactive CMKLR1 reported, we used a structural model of inactive CMKLR1 predicted using AlphaFold2 (AF structure) [44–47] in our structural comparison analysis (**Fig 3A**). Compared to the AF structure of inactive CMKLR1, there are inward rearrangements of TM5 and TM7 and an outward displacement of TM6 at the cytoplasmic region of $G_i$-coupled CMKLR1 (**Fig 3B**), which are characteristic of the active conformations of Class A GPCRs [48–51]. Noticeably, the approximately 4.5-Å displacement of the cytoplasmic end of TM6 is much less prominent compared to those observed in the activation of other $G_i$-coupled receptors such as μ-opioid receptor (approximately 10 Å) [38], the melatonin receptor MT1 (approximately 15 Å), adenosine $A_1$ receptor (approximately 10.5 Å) [52], and rhodopsin (approximately 10 Å) [53]. This may suggest a relatively smaller energy barrier for CMKLR1 to change from the inactive to the fully active conformation.

Alignment of the $G_i$-coupled active structure of CMKLR1 to the AF structure showed very subtle structural differences at the extracellular region including residues in the ligand-binding pocket (**Fig 3C**). This implies that the ligand-binding pocket of apo CMKLR1 is relatively rigid and well poised for chemerin binding. Nevertheless, at the bottom region of the binding pocket, Ser157 and Phe156 of chemerin9 cause conformational changes of N116$^{3.29}$ and L119$^{3.32}$ in TM3 due to direct steric effects, which further lead to an outward shift and rotation of TM3 (**Fig 3D**). As a result, M123$^{3.36}$ shifts towards W273$^{6.48}$, forcing it to move towards and inducing significant displacement of F269$^{6.44}$ (**Fig 3D**). For rhodopsin-like Class A GPCRs, W$^{6.48}$ and F$^{6.44}$ constitute a conserved "activation switch" microdomain. Rearrangement of this microdomain links extracellular agonist-binding to cytoplasmic conformational changes in receptor activation [49,50,54,55]. Indeed, the movement of W273$^{6.48}$ and F269$^{6.44}$ in CMKLR1 breaks the continuous helical structure of TM6, resulting in an outward displacement of its cytoplasmic segment (**Fig 3D**), a hallmark of GPCR activation. W273$^{6.48}$ is also important for the correct folding of the receptor. Mutations at this residue to Ala, Tyr, or Leu disrupted receptor membrane localization and caused the loss of activity of chemerin9 (**S4A and S4B Fig**). It is to be noted that the AlphaFold-predicted inactive GPCR structures may not be entirely precise. Our proposed CMKLR1 activation mechanism, which involves

conformational changes of Ser157 and Phe156 of chemerin9 and N116$^{3.29}$ and L119$^{3.32}$ of CMKLR1, only represents a plausible hypothesis.

## Critical molecular features for the action of chemerin9 revealed by mutagenesis studies

To investigate the critical molecular features of the chemerin9-CMKLR1-G$_i$ complex structure in receptor activation, we conducted mutagenesis studies using Ca$^{2+}$ flux assays. First, we tested 4 residues of CMKLR1, Y47$^{1.39}$, N191$^{4.77}$, S295$^{7.32}$, and L298$^{7.35}$ by exchanging them to Ala (**S4E Fig**). These residues are close to, but do not directly interact with, chemerin9 in our structure (**S4D FIg**). All 4 mutants CMKLR1 variants have been detected in the cell membrane (**S4A Fig**), and we did not detect significant changes in the chemerin9 activation for these receptor variants (**S4E Fig**), suggesting a minor role of these residues in the action of chemerin9. We then tested a series of mutations of CMKLR1 residues involved in the interactions with chemerin9 in our structure. All mutants displayed cell surface expression with the exception of W273$^{6.48}$ variants, Y276$^{6.51}$A, H95$^{2.60}$A, and H95$^{2.60}$L. The F88$^{2.53}$A and Y47$^{1.39}$A variants are only partially expressed in the membrane (**S4A and S5 Figs**).

At the extracellular region of CMKLR1, the first N-terminal residue of chemerin9, Tyr149, likely engages in a weak π–π interaction [56] with F294$^{7.31}$ of CMKLR1 (**Fig 4A**). Mutations of F294$^{7.31}$ to Ala or His resulted in a decrease in the potency of chemerin9, while the F294$^{7.31}$L

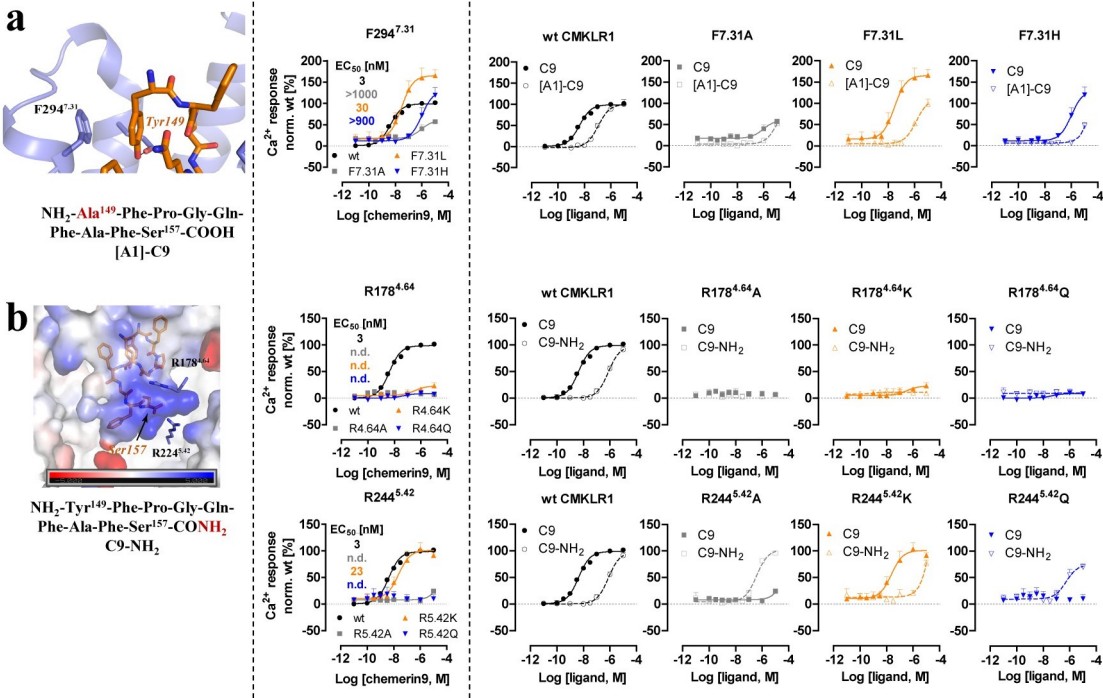

**Fig 4. Mutagenesis studies focused on the CMKLR1 interactions mediated by the C- and N-terminus of chemerin9. (a)** The structure of the chemerin9 (C9, orange) residue Tyr149 at N-terminus and the sequence of [A1]-chemerin9 ([A1]-C9) are shown in the left panel. Ca$^{2+}$ signal transduction of wild-type CMKLR1 and receptor variants stimulated with chemerin9 is shown in the middle panel. Double cycle mutant analysis with [A1]-C9 is shown in the right panel. **(b)** The structure of the chemerin9 residue Ser157 at the C-terminus and the sequence of chemerin9-NH$_2$ (C9-NH$_2$) are shown in the left panel. Ca$^{2+}$ signal transduction of wild-type CMKLR1 and receptor variants stimulated with chemerin9 is shown in the middle panel. Double cycle mutant analysis with C9-NH$_2$ is shown in the right panel. All Ca$^{2+}$ assay measurements were performed with transiently transfected HEK293 in triplicates and at least 3 times. Data points demonstrate mean ± SEM. Data analysis was performed with GraphPad Prism 5.0. The underlying data for Fig 4A and 4B can be found in S2 Data.

mutation increased the maximum $Ca^{2+}$ response. These findings suggest that a hydrophobic residue at the 7.31 position of CMKLR1 is crucial in maintaining the side chain of Tyr149 of chemerin9 in the appropriate position for the binding of this peptide agonist. In addition, a new chemerin9 peptide with the Tyr149Ala mutation ([A1]-C9 peptide) (**Table 2**) showed reduced potency for the wt receptor or the 3 CMKLR1 variants F294[7.31]A, F294[7.31]L, and F294[7.31]H, likely due to the elimination of the hydrophobic interaction between the agonist and the 7.31 residues of the receptors. The double-mutant cycle analysis indicates that Tyr149 of chemerin9 is not exclusively interacting with F294[7.31] of CMKLR1 [57].

At the C-terminal end of chemerin9, which is bound at the bottom of the binding pocket, the carboxylate group of the last residue Ser157 is located in a highly positively charged environment surrounded by CMKLR1 residues R178[4.64] and R224[5.42] (**Fig 4B**). Mutations of either arginine to Ala or Gln resulted in undetec signaling activity of CMKLR1 induced by chemerin9, indicating the crucial role of the positive charge environment around Ser157 for ligand binding. Interestingly, the mutation R178[4.64]K, but not R224[5.42]K, caused the loss of CMKLR1 signaling induced by chemerin9, suggesting that the length of the side chain of R178[4.64], not only its positive charge, is also essential to ligand binding. To further examine the charge interactions between Ser157 of chemerin9 and CMKLR1 in ligand binding (**Fig 4B**), 2 chemerin9 derivatives were generated: one mutating Ser157 to Ala ([A9]-C9) and the other changing the carboxylate group of Ser157 to carboxamide (C9-$NH_2$) (**S4C Fig** and **Table 2**). While the carboxamide group in C9-$NH_2$ is likely to be positively charged, [A9]-C9 still maintains the last negatively charged carboxylate group. Compared to [A9]-C9, C9-$NH_2$ caused a more significant decrease in potency, indicating that the negative charge of the C-terminal end of chemerin9 is critical to ligand binding (**S4C Fig**). Moreover, while C9-$NH_2$ showed no activity on the 3 CMKLR1 variants, R178[4.64]A, R178[4.64]K, and R178[4.64]Q, it partially rescued the loss of activity of chemerin9 on the CMKLR1 R224[5.42]A and R224[5.42]Q variants (**Fig 4B**). It is possible that the salt bridge between the C-terminal carboxylate of chemerin9 and R224[5.42] of CMKLR1 is essential for maintaining the ligand in the right conformation for activating the receptor. For C9-$NH_2$, the carboxamide group at the C-terminus may form polar interactions with other nearby residues in the CMKLR1 R224[5.42]A and R224[5.42]Q variants, thereby stabilizing the carboxamide group in the correct position. However, in the wt CMKLR1 or the R224[5.42]K variant, the same-charge repulsion may cause this group of C9-$NH_2$ to swing away from R224 (wt) or K224 (variant), resulting in the significantly lower activity of C9-$NH_2$ on both receptors.

Phe156 of chemerin9 plays a crucial role in the interactions with the receptor, forming hydrophobic or π–π interactions with CMKLR1 residues F88[2.53], H95[2.60], M123[3.36], Y276[6.51], and I306[7.43] (**Figs 2E and 5**). Consistently, when Phe156 was mutated to Ala in chemerin9, the resulting peptide ([A8]-C9) (**Table 2**) lost nearly all activity on CMKLR1 (**Fig 5**). We also examined the impact of mutations at 3 aromatic residues in CMKLR1, F88[2.53], H95[2.60], and Y276[6.51], on ligand binding. Mutating F88[2.53] to Ala decreased the potency of chemerin9 by approximately 100-fold, but changing it to Leu or His led to a more significant decrease in agonist potency. These results suggest that neither the aromaticity nor the size of F88[2.53] is important for the action of chemerin9. Furthermore, there was a gain of potency of [A8]-C9 for the F88[2.53]A and F88[2.53]L variants compared to the wt CMKLR1. Therefore, while F88[2.53] is important in chemerin9 binding, the role of this residue appears to be complex. In the case of H95[2.60], mutation to Ala, Leu, or Lys reduced the potency of chemerin9 to different extents, possibly by altering its interaction with Phe156 of chemerin9. For Y276[6.51], exchange by Ala completely abolished the activity of chemerin9, likely due to poor cell surface expression of the receptor variant (**S5 Fig**). Interestingly, when mutated it to Leu, the potency of chemerin9 decreased by 10-fold, while mutation to Phe increased the maximum activity of chemerin9

**Table 2. Peptide sequences.**

| peptide | | sequence | avg. MW [g/mol] | exact mass [g/mol] | purity (HPLC) |
|---|---|---|---|---|---|
| chemerin9 | C9 | YFPGQFAFS | 1,063.18 | 1,062.48 | >98%[1,2] |
| [A1]-chemerin9 | [A1]-C9 | AFPGQFAFS | 971.08 | 970.45 | >95%[1,3] |
| [A8]-chemerin9 | [A8]-C9 | YFPGQFAAS | 987.08 | 986.45 | >95%[1,3] |
| chemerin9-NH$_2$ | C9-NH$_2$ | YFPGQFAFS-NH$_2$ | 1,062.19 | 1,061.50 | >95%[2,3] |

Peptides were synthesized by solid phase peptide synthesis Fmoc/*t*Bu strategy.

Squared brackets symbolize exchanges within in the chemerin9 structure.

Purity was investigated by different HPLC columns

[1]Phenomenex Jupiter Peptide Proteo C-12, 250 × 4.6 mm, 90 Å, 4 µm.

[2]Phenomenex Aeris Peptide XB-C18, 250 × 4.6 mm, 100 Å, 3.6 µm.

[3]Phenomenex Kinetex Peptide biphenyl, 250 × 4.6 mm, 100 Å, 5 µm.

MW, molecular weight.

(**Fig 5**), indicating the importance of the aromaticity of Y276$^{6.51}$ for the activity of chemerin9. None of the mutations of H95$^{2.60}$ or Y276$^{6.51}$ rescued the activity of [A8]-C9.

## Conformational dynamics of the chemerin9-CMKLR1-G$_i$ complex

Chemerin9 is a potent peptide agonist of CMKLR1. To better understand the conformational dynamics of the receptor and chemerin9 in G$_i$ coupling, we performed 5 different sets of MD

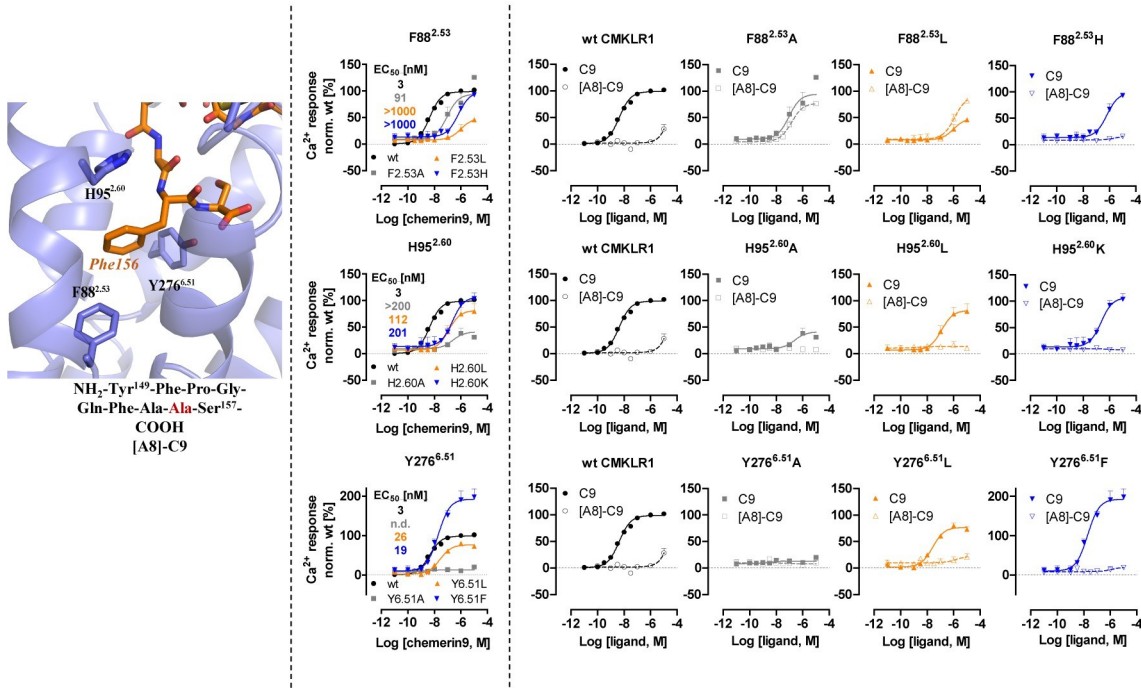

**Fig 5. Mutagenesis studies focused on the CMKLR1 interactions mediated by Phe156 of chemerin9.** The structure of the C9 (orange) residue Phe156 and the sequence of [A8]-C9 peptide are shown in the left panel. Ca$^{2+}$ signal transduction of wt CMKLR1 and receptor variants stimulated with C9 is shown in the middle panel. Double cycle mutant analysis with [A8]-C9 and receptor variants is shown in the right panel. The Ca$^{2+}$ assays were performed with transiently transfected HEK293 were performed with at least 3 independent experiments in triplicates. Data points demonstrate mean ± SEM. The underlying data for Fig 5 can be found in S3 Data. CMKLR1, chemokine-like receptor 1; C9, chemerin9; wt, wild-type.

simulations of the CMKLR1-G$_i$ complex or CMKLR in the absence or presence of chemerin9 (see details in **Methods**). Each system was subjected to 3 runs of 100-ns MD simulations, resulting in a total of 1.5 microseconds of simulation data. The CMKLR1-G$_i$ complex remained stably associated throughout all simulation runs with or without chemerin9. Noticeably, we observed that CMKLR1 exhibited slightly more conformational fluctuations in the absence of chemerin9, despite being coupled and stabilized by G$_i$ (**S6A Fig**). These results suggest that chemerin9 may further stabilize the active conformation of CMKLR1 even in the presence of G$_i$.

As mentioned above, in our cryo-EM structure, the carboxylate group of Ser157 is located in a positively charged environment (**Fig 4B**). To investigate the impact of charges of chemerin9 on ligand binding, we conducted simulations of CMKLR1 bound to chemerin9 with both neutral and charged amino- and carboxyl-termini (N-ter and C-ter). Both charged and noncharged chemerin9 remained stably bound in the binding pocket throughout the simulations, indicating that the binding is overall stabilized by a network of interactions rather than specific pairs. Yet, to further evaluate the binding energy of chemerin9, we used the Molecular Mechanics/Generalized Born Surface Area (*MM/GBSA*) method [58], as per previous protocol [59] (**Fig 6A**). Our calculations showed that charged chemerin9 exhibited a lower $\Delta H_{MM/GBSA}$ of binding, indicating a higher affinity for CMKLR1 compared to noncharged chemerin9 (**Fig 6A**). We attributed this increase in binding affinity primarily to the charged carboxyl termini of chemerin9. This is consistent with our mutagenesis studies suggesting the important role of

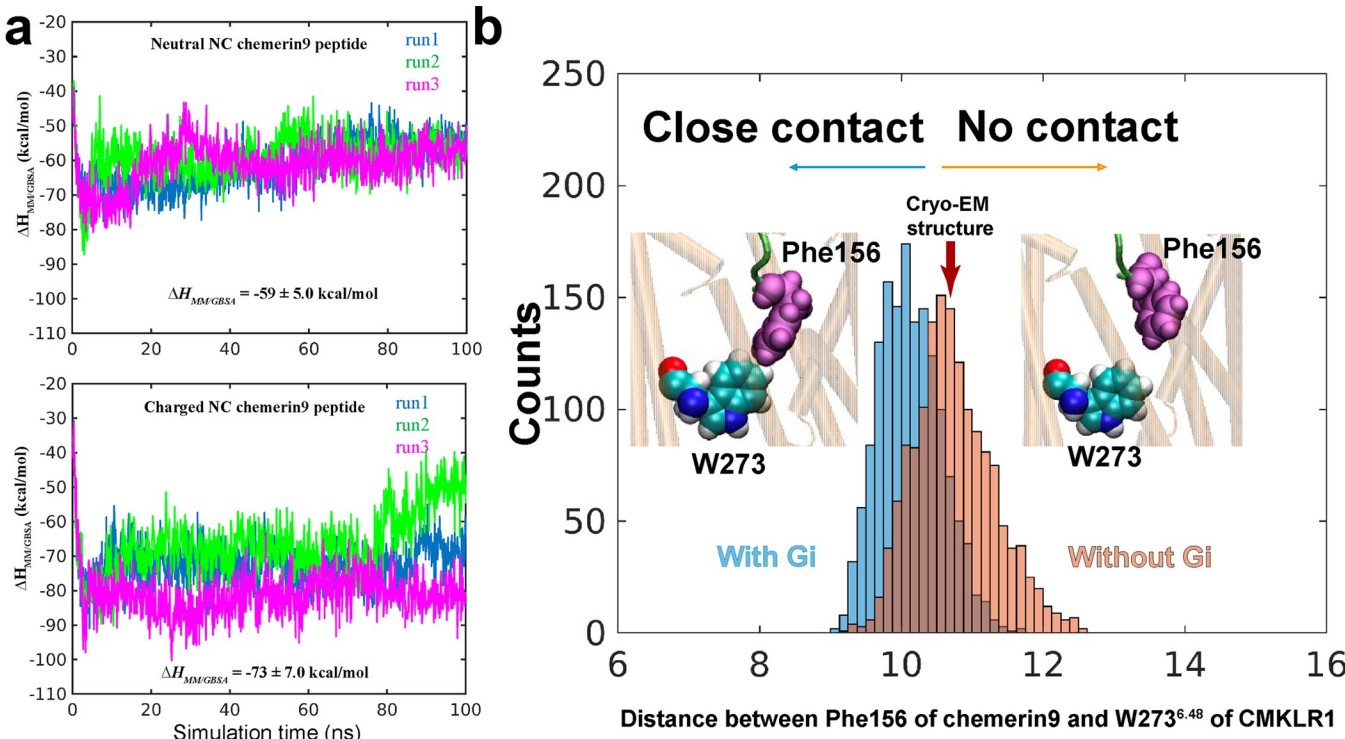

**Fig 6. Binding of chemerin9 probed by MD simulations.** (**a**) MM/GBSA calculations (multiple runs of 100 ns each) indicated the more favorable interactions achieved in the presence of charged amino acids at the C- and N-termini of chemerin9 (lower panel), compared to those in the presence of neutral amino acids (upper panel). (**b**) Gi-bound receptor exhibits closer association with chemerin9, evidenced by the closer distance between Phe156 (chemerin9) and W273$^{6.48}$ (CMKLR1) observed in the runs conducted with (cyan histogram) and without (orange histogram) Gi protein bound to the receptor. The underlying data for Fig 6A and 6B can be found in S4 Data. CMKLR1, chemokine-like receptor 1; MD, molecular dynamics; MM/GBSA, Molecular Mechanics/Generalized Born Surface Area.

charge–charge interactions between the C-terminal carboxylate group of chemerin9 and CMKLR1.

During our simulations of CMKLR1 alone with charged chemerin9, we observed chemerin9 undergoing up and down movements within the binding pocket. However, in the presence of $G_i$, which stabilized the active conformation of CMKLR1, chemerin9 gradually inserted into the pocket and moved down towards W273[6.48] (**Fig 6B**). This observation suggests that the active CMKLR1 conformation favors a close distance between the peptide agonist and the transmission switch motif. It is possible that this state only represents a local minimum energy state, which was not captured by the cryo-EM structure. These findings offer new insights into how the coupled conformational dynamics of the receptor and the peptide agonist underlie receptor activation and signaling.

## Molecular details of $G_i$-coupling to CMKLR1

The overall $G_i$-coupling interaction profile in our structure of $G_i$-coupled CMKLR1 is highly similar to that of $G_i$-coupled FPR1 and FPR2. The C-terminal α5 of $Gα_i$ in the $G_i$-coupled CMKLR1 and FPR2, which is the major interaction site between $G_i$ and receptors, can be well superimposed if the 2 receptors are aligned, suggesting a highly conserved $G_i$-coupling mechanism (**Fig 7A**).

In the cytoplasmic cavity of CMKLR1, hydrophobic side chains of residues I344, L348, L353, and F354 of $Gα_i$ on one side of α5 form hydrophobic interactions with CMKLR1 residues I243[5.61], L247[5.65], L252[ICL3], P258[6.33], and I261[6.36] (**Fig 7B**). The side chain of CMKLR1 R137[3.50] in the conserved DR[3.50]Y/F/C motif forms a hydrogen bond with the main chain carbonyl of C351 of $Gα_i$, which is also observed in many other GPCR-$G_i$ complexes (**Fig 7B**). Another polar interaction is observed between the side chain of N347 of $Gα_i$ and the main chain carbonyl of S140[3.53] of CMKLR1 (**Fig 7B**). In addition to those interactions within the cytoplasmic cavity of CMKLR1, V145 in ICL2 of CMKLR1 forms hydrophobic interactions with L194 in β2-β3 loop and F336 in α5 of $Gα_i$, and N149 in ICL2 of CMKLR1 forms a hydrogen bond with the main chain carbonyl of D193 of $Gα_i$ (**Fig 7C**). No direct interactions are observed between the receptor and the $G_{βγ}$ subunits.

During our MD simulations of the CMKLR1-$G_i$ complex, the $G_i$ heterotrimer remained stably bound to the receptor, regardless of the presence of chemerin9, across each 100-ns

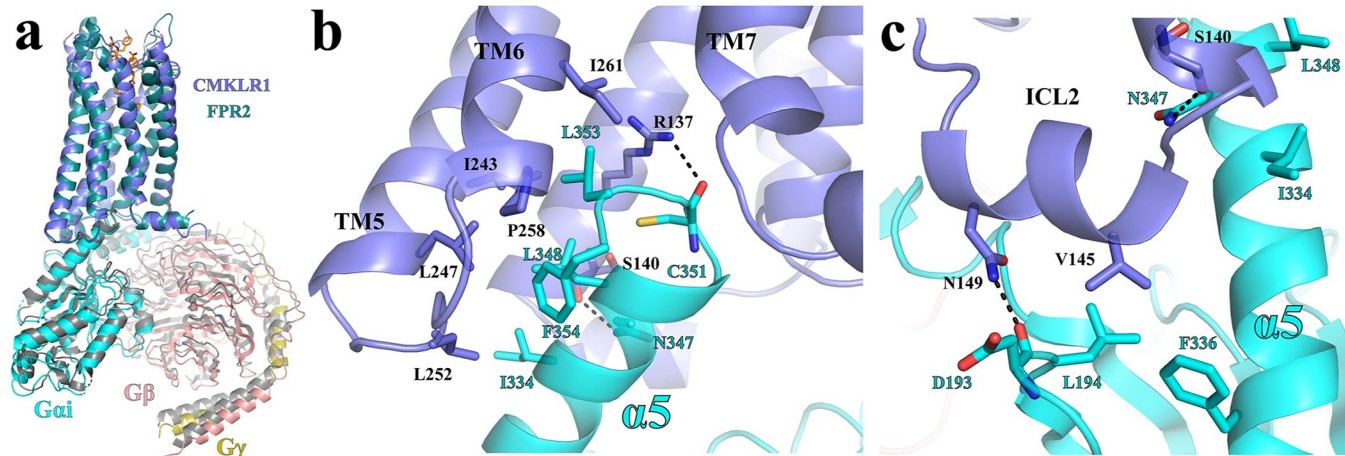

**Fig 7. $G_i$-coupling to CMKLR1. (a)** Structural alignment of the FPR2-Gi and CMKLR1-Gi complexes. **(b)** Interactions between CMKLR1 (blue) and the α5 of $Gα_i$ (cyan). **(b)** Interactions between the ICL2 of CMKLR1 (blue) and $Gα_i$ (cyan).

simulation run (**S6A Fig**). We note that such a time frame may not be long enough to capture the potential conformational transitions of Gi heterodimer during activation and dissociation. The hydrophobic interactions identified in our cryo-EM structure remained consistent throughout the simulations. Notably, we observed several intermittent interfacial salt bridges between CMKLR1 and $G_i$ (**S6B Fig**), such as those between D312 of Gβ and K68 and K70 in ICL1 of CMKLR1, between D315 and E318 of Gα$_i$ and K254 in ICL3 of CMKLR1, and between D261 of Gα$_i$ and K325 in helix8 of CMKLR1 (**S6B Fig**). Interestingly, a network of hydrogen bonds and salt bridges was formed between D350 of Gα$_i$ and R137 and R151 of CMKLR1 (**S5B Fig**). This highlights the role of charge–charge interactions in the association of $G_i$ with CMKLR1.

## Discussion

The signaling pathway mediated by CMKLR1 has been reported to exhibit a dual role in inflammation, with both pro- and anti-inflammatory effects as observed in previous studies [7,9]. The functional outcome of CMKLR1 signaling may differ based on the stage of inflammation. Notably, recent research has highlighted the importance of the pro-resolving lipid ligand RvE [21–25] in mediating much of the anti-inflammatory function of CMKLR1 during the resolution phase of inflammation [60]. Our investigation using primary human macrophages revealed that chemerin9 induces a macrophage phenotype that is distinct from the simplified M1 and M2 classification of pro- and anti-inflammatory phenotypes, indicating a more complex role for CMKLR1 signaling in modulating macrophage function. In addition, our structure of the CMKLR1-$G_i$ complex bound to chemerin9 reveals an "S-shape" binding mode of chemerin9 and the molecular determinants for its agonistic action. Together with mutagenesis and MD simulations studies, our structure provides insights into the mechanism of how chemerin9 activates CMKLR1 to induce $G_i$ signaling.

The development of drugs targeting CMKLR1 has been limited, with only a few small molecule antagonists reported to date. However, a recent study has identified an agonist antibody of CMKLR1 that showed promise in inducing chronic resolution of inflammation in animal models with chronic colitis [31]. This promising proof-of-concept study indicates that small molecule CMKLR1 agonists may also exhibit potential anti-inflammatory therapeutic effects, which have yet to be realized. FPR2, a closely related chemoattractant GPCR, has also been suggested to promote pro-resolving signaling [11,12]. Unlike CMKLR1, numerous small molecule agonists of FPR2 have been developed, with some currently under clinical investigation [61–64]. Previous studies from us have shown that a small molecule agonist of FPR2 shares the same binding site as peptide agonists in the bottom region of the ligand-binding pocket [36,37]. Our structure of CMKLR1 revealed the molecular details of the ligand-binding pocket of CMKLR1 and critical molecular determinants for the agonistic action of chemerin9. A previous modelling study suggested a "U-shape" binding mode of chemerin9 based on comparison of cyclic and linear chemerin9 peptide variants [65]. In our structure, chemerin9 adopts an "S-shape" binding mode as it in the recently published structure of the CMKLR1-$G_i$-chemerin9 complex [66]. In fact, the 2 independently determined structures exhibit a high degree of superimposition, particularly for the receptor and chemerin9, thereby providing further validation for our structure (**S7 Fig**). It is conceivable to design small molecular agonists of CMKLR1 that mimic the binding of chemerin9 in our structure as novel pro-resolving or anti-inflammatory therapeutics. In addition, our structure suggested a potential lipid-binding site above ICL2 (**S3B Fig**), which could serve as the allosteric binding site for RvE1. Interestingly, a number of allosteric modulators for C5aR, a close phylogenetic neighbor of CMKLR1, including avacopan, a drug for treating an autoimmune disease, target a similar site right above ICL2

of C5aR [41]. Therefore, the development of small molecules as allosteric modulators of CMKLR1 targeting this site may also be another strategy to modulate CMKLR1 signaling in order to control inflammation.

## Materials and methods

### Protein expression and purification

The coding sequence of wt human CMKLR1 was cloned into pFastBac vector with a signal peptide followed by a FLAG tag and a peptide sequence corresponding to the N-terminal fragment of the human $\beta_2$-adrenergic receptor, which is to facilitate protein expression in Sf9 cells, before the N-terminal end and a C-terminal 8xHis tag. For the $G_i$ protein, a dominant negative form of $G\alpha_{i1}$ (DNG$\alpha_{i1}$) with 4 mutations (S47N, G203A, E245A, and A326S) was constructed to reduce the nucleotide binding. Human DNG$\alpha_{i1}$, human G$\beta_1$ with an N-terminal His$_6$-tag, human G$\gamma_2$, and a single-chain antibody scFv16 were cloned into pFastBac vector.

CMKLR1, DNG$\alpha_{i1}$, and G$\beta_1\gamma_2$ were coexpressed in Sf9 insect cells using Bac-to-Bac method. Sf9 cells were infected with 3 types of viruses at the ratio of 1:1:1 for 48 hours at 27˚C. After infection, cell pellets were harvested and stored at −80˚C. Cell pellets were thawed in lysis buffer containing 20 mM HEPES (pH 7.5), 50 mM NaCl, 10 mM MgCl$_2$, 5 mM CaCl$_2$, 2.5 µg/ml leupeptin, 300 µg/ml benzamidine. To facilitate complex formation, 1 µM chemerin9, 25 mU/ml Apyrase (NEB), and 100 µM TCEP was added and incubated at room temperature for 2 hours. The cell membranes were isolated by centrifugation at 25,000 × g for 40 minutes and then resuspended in solubilization buffer containing 20 mM HEPES (pH 7.5), 100 mM NaCl, 0.5% (w/v) lauryl maltose neopentylglycol (LMNG, Anatrace), 0.1% (w/v) cholesteryl hemisuccinate (CHS, Anatrace), 10% (v/v) glycerol, 10 mM MgCl$_2$, 5 mM CaCl$_2$, 12.5 mU/ml Apyrase, 1 µM chemerin9, 2.5 µg/ml leupeptin, 300 µg/ml benzamidine, 100 µM TECP at 4˚C for 2 hours. The supernatant was collected by centrifugation at 25,000g for 40 minutes and incubated with nickel Sepharose resin (GE Healthcare) at 4˚C overnight. The resin was washed with a buffer A containing 20 mM HEPES (pH 7.5), 100 mM NaCl, 0.05% (w/v) LMNG, 0.01% (w/v) CHS, 20 mM imidazole, and 1 µM chemerin9, 2.5 µg/ml leupeptin, 300 µg/ml benzamidine, 100 µM TECP. The complex was eluted with buffer A containing 400 mM imidazole. The eluate was supplemented with 2 mM CaCl$_2$ and loaded onto anti-Flag M1 antibody resin. After wash, the complex was eluted in buffer A containing 5 mM EDTA and 200 µg/ml FLAG peptide and concentrated using an Amicon Ultra Centrifugal Filter (MWCO, 100 kDa). Finally, a 1.3 molar excess of scFv16 was added to the elution. The sample was then loaded onto a Superdex 200 Increase 10/300 column (GE Healthcare) preequilibrated with buffer containing 20 mM HEPES (pH 7.5), 100 mM NaCl, 0.00075% (w/v) LMNG, 0.00025% (w/v) GDN, 0.00015% (w/v) CHS, 1 µM chemerin9, and 100 µM TECP. Peak fractions of the complex were collected and concentrated to 5 mg/ml for cryo-EM experiments.

scFv16 was expressed in Hi5 insect cells using Bac-to-Bac expression system. To purify the protein, the cell supernatant was collected and purified by nickel affinity chromatography before the C-terminal His$_8$-tag was removed by TEV protease. The protein was further purified by size exclusion chromatography using a Superdex 200 Increase 100/300 GL column (GE Healthcare). The purified scFv16 were concentrated, flash-frozen by liquid nitrogen and stored at −80˚C.

### cryo-EM sample preparation and data collection

For cryo-EM grids preparation of chemerin9-CMKLR1-$G_i$ complex, 3 µl protein sample was applied to 1.2/1.3 UltrAufoil 300 mesh grids that had been plasma cleaned and plunged into liquid ethane using an FEI Vitrobot Mark IV (Thermo Fisher Scientific). Approximately 3,992

movies were collected using Titan Krios transmission electron microscope, equipped with a Gatan K3 Summit direct electron detector and an energy filter. Images were recorded with a nominal magnification of 105,000× using the SerialEM [67] software in super-resolution mode with a calibrated pixel size of 0.848 Å and a defocus range of −1.0 to −1.8 μm. Each stack was dose fractionated to 52 frames with a total dose of 56 e$^-$/Å$^2$.

## cryo-EM structure determination

Image stacks were subjected to patch motion correction using cryoSPARC [68]. Contrast transfer function (CTF) parameters were calculated using the patch CTF estimation tool. Total of 3,900,728 particles of chemerin9-CMKLR1-$G_i$ complex were autopicked and then subjected to 2D classification to discard poorly defined particles. After ab initio reconstruction and heterogeneous refinement, 242,745 particles were subjected to nonuniform refinement and local refinement, which generated a map with an indicated global resolution of 2.94 Å at a Fourier shell correlation (FSC) of 0.143. Local resolution was estimated in cryoSPARC [68].

The AlphaFold-predicted structure of CMKLR1 and structures of $G_i$ and scFv16 obtained from the FPR1-$G_i$-scFv16 complex (PDB ID 7T6T) were used as initial models for docking into the cryo-EM map using Chimera [69]. The structure of chemerin9-CMKLR1-$G_i$ was subsequently generating using iterative manual building and adjustment in Coot [70], followed by real-space refinement in Phenix [71]. The final model were validated by Molprobity [72]. Detailed statistics for data collection, processing, and structure refinement statistics are provided in the **Table 1**.

## Macrophage differentiation assays

To generate human peripheral blood-derived macrophages, buffy coat was isolated from leukapheresis cones by Ficoll–Paque centrifugation. Monocytes were enriched from human PBMC by CD14 beads positive selection (Miltenyi Biotec) and cultured in IMDM supplanted with 10% human AB serum (Omega Scientific, Cat# HS-20) for 7 days for them to differentiate to macrophages.

Differentiated macrophages were then replated to low-bind plate at 0.15 million cells per 24-well plate, in 500 μl medium overnight before stimulated with various condition. Human macrophage medium was used as M0 reference culture condition, while 10 ng/ml IFNg plus 10 ng/ml LPS supplement were used as M1 reference, and 10 ng/ml IL-10 supplement used as M2 reference. Cells were stimulated with 10 ng/mL chemerin9 for 72 hours, before stained with FITC anti-human CD11b (M1/70), PE anti-human HLA-DR (L243), BV510 anti-human CD86 (FUN-1), AF647 anti-human CD206 (15–2), and APC.Cy7 anti-human CD163 (GHI/61). Data were acquired on Cytek Aurora flowcytometry, analysis on Flowjo. MFI quantifications were normalized to M0 condition and statistically analyzed by 2-way ANOVA, with Dunnett's multiple comparisons test; all groups were pair analyzed with M0 group as reference.

## Peptide synthesis

In general, all used peptides were synthesized by an orthogonal 9-fluorenylmethoxycarbonyl/*tert*-butyl (Fmoc/*t*Bu) solid phase peptide synthesis (SPPS) strategy. The synthesis was performed on a Syro II peptide synthesizer (MultiSynTech, Bochum, Germany) on a scale of 15 μmol per resin. A Wang-resin, which was preloaded with serine for chemerin9 (C9) was used for automatic SPPS. The coupling was performed twice with 8 equivalent of the respective, Fmoc-protected amino acid activated in situ with equimolar amounts of oxyma and DIC in DMF for 30 minutes. By incubation with 40% piperidine in DMF (v/v) for at least 3 minutes

and 20% piperidine in DMF (v/v) for 10 minutes, Fmoc was deprotected. Peptides were cleaved from the resin by incubation with 90% TFA and the following scavengers: 7% thioanisole, 3% ethanedithiol (v/v/v) for 3 hours at room temperature and precipitated in ice-cold diethyl etherate −20°C for at least 3 hours, washed with diethyl ether.

Peptides were purified on a preparative RP-HPLC: phenomenex Aeris Peptide XB-C18, 250 × 21.2 mm, 100 Å pore size, and 5 μm particle size (Phenomenex, Torrence, USA). A linear gradient of eluent B (0.08% TFA in ACN (v/v)) in eluent A (0.1% TFA in $H_2O$ (v/v)) was used on the column for all RP-HPLCs. For the preparative RP-HPLC, a gradient of 20% to 50% in 30 minutes of eluent B in A was used. Only peptides fractions with a purity of >95% were combined, and the purity was determined by RP-HPLC on a Jupiter 4 μm Proteo 90 Å C12 (Phenomenex), an Aeris 3.6 μm 100 Å XB-C18 (Phenomenex) or on a Kinetex Peptide biphenyl, 250 × 4.6 mm, 100 Å, 5 μm (Phenomenex) column (Table 1). The identity of all peptides were confirmed by MALDI-ToF MS on an Ultraflex II and ESI-MS (Bruker Daltonics, Billerica, USA).

## Mutagenesis of CMKLR1

The receptor construct hCMKLR1b-eYFP in pVitro2 vector was used as the wild type and template for the mutations. The variants were named according to the nomenclature of Ballesteros and Weinstein [43]. The receptor constructs encode the corresponding CMKLR1 sequence with a C-terminally attached enhanced yellow fluorescent protein (eYFP). Single-amino acid exchanges were generated by Q5 site-directed mutagenesis (New England Biolabs GmbH, Frankfurt am Main) using the appropriate primer pairs that were designed by using NEBaseChanger. Sanger sequencing confirmed the identity of the constructs performed by Microsynth Seqlab GmbH.

## $Ca^{2+}$ flux assays

HEK293 cells were transfected with the wt CMKLR1 or variant plasmid and the chimeric G protein GαΔ6qi4myr overnight using Metafectene Pro according to the manufacturer's protocol. The vector for the chimeric GαΔ6qi4myr protein was kindly provided by E. Kostenis (Rheinische Friedrich-Wilhelms-Universität, Bonn, Germany) [73]. A black 96-well plate with μClear bottom was coated with 0.001% poly-D-lysine in DPBS (v/v) for 10 minutes and dried overnight. Transfected cells were seeded into these coated 96-well plates (100,000 cells/wells) and incubated overnight at standard conditions (5% $CO_2$, 95% humidity, 37°C). On the following day, the $Ca^{2+}$-flux assay was performed in assay buffer (20 mM HEPES, 2.5 mM Probenecid in HBSS (pH 7.5)). Cells were incubated with Fluo-2-AM solution (0.3% (v/v) and Pluronic-F127 (0.3% (v/v) in assay buffer. After 1 hour of incubation at 37°C, the dye solution was replaced by 100 μl assay buffer. The basal $Ca^{2+}$ level was recorded for 20 seconds with a Flexstation 3 (Molecular Devices, λex = 485 nm, λem = 525 nm). The ligand was added afterwards, and the $Ca^{2+}$ response was measured for another 40 seconds. The resulting maximum over basal value was calculated for each well and normalized to the top and bottom values of the control containing the wt CMKLR1 stimulated by the respective peptide. All experiments were performed in triplicates, and each experiment was repeated at least 3 times. Nonlinear regression was calculated using GraphPad Prism 5.

## Measuring protein expression on cell surface

For determining the receptor variant localization, microscopy studies were performed at a confocal microscope (ApoTome, 63×/1.40 oil objective, Carl Zeiss) making use of the auto-fluorescence of eYFP. Therefore, 150,000 cells/200 μl were seeded into each well of an 8-well

Ibidi μ-Slide and incubated at 37˚C for over 30 hours. The next day, 1,000 ng of the variant was transiently transfected in OptiMEM without FBS by using Lipofectamine 2000 according to the protocol from the vendor. After 1 hour, the transfection reagents were removed and replaced by normal culture medium containing FBS. Cells were incubated 1 day at standard conditions. The medium was removed and replaced by 200 μl OptiMEM and 1 μl Hoechst incubated for 30 minutes at 37˚C. Microscopy studies were performed at least 2 times, and one representative is shown in results.

## Molecular dynamics (MD) simulations

**MD simulation systems and protocol.** All MD simulation systems were prepared using CHARMM-GUI [74], based on the resolved chemerin9-CMKLR1-$G_i$ complex. For the chemerin9 peptide ($Y^{149}$FPGQFAFS$^{157}$), we adopted 2 different conformers: (i) the N- and C-termini were charged (charged chemerin9) as ($H_2^+$-Tyr-Phe-Pro-Gly-Gln-Phe-Ala-Phe-Ser-$O^-$); and (ii) the N-and C-termini were neutral (neutral chemerin9), in which the terminal group were taken as (H-Tyr-Phe-Pro-Gly-Gln-Phe-Ala-Phe-Ser-OH). Five distinct MD simulation systems were constructed: (1) CMKLR1- $G_i$ complex bound to charged chemerin9; (2) CMKLR1-$G_i$ complex bound to neutral chemerin9; (3) *Apo* CMKLR1-$G_i$ complex, in which the bound peptide was removed; (4) *Apo* CMKLR1, in which both $G_i$ protein and the bound peptide were removed; and (5) CMKLR1 bound to charged chemerin9, in which the $G_i$ protein was removed. For each system, the protein complex was first oriented using the PPM webserver [75] and then was embedded into membrane/lipids composed of 1-palmitoyl-2-oleoyl-sn-glycero-3-phosphocholine (POPC), using CHARMM-GUI *Membrane Builder* module [76], and the resolved cholesterol (CHOL) molecules were included. Fully equilibrated transferable intermolecular potential 3P (TIP3P) waters were added to build a simulation box of $132 \times 132 \times 165$ Å$^3$ or $132 \times 132 \times 111$ Å$^3$ for the CMKLR1-Gi complex or CMKLR1 protein, respectively; $Na^+$ and $Cl^-$ ions were added to obtain a 0.15 M NaCl neutral solution. For each system, 3 runs of 100 ns MD simulations (a total of 1.5 microseconds) were performed using NAMD [77], following the well-established/default protocol [78,79]. VMD [80] with in-house scripts were used for visualization and trajectory analysis.

**MM/GBSA computational protocol and parameters.** Peptide Chemerin9 binding energies were evaluated using the *MM/GBSA* method [58], adopted from previous protocol [59]. First, we took our MD trajectories generated for the chemerin9-CMKLR1 complex, in explicit solvent. Second, all solvent molecules, ions, lipids, or $G_i$ protein if any, were removed from each MD snapshot, yielding trajectories for CMKLR1, chemerin9 and chemerin9-CMKLR1 complex. Third, for each of these trajectories, the MM/GBSA free energy was calculated using *GBIS* module [81] implemented in NAMD. The binding free energy change $\Delta H_{MM/GBSA}$ was calculated following previous protocol [59] with the default parameters set in the *GBIS* module [81]. Briefly, CHARMM36 force field with CMAP corrections [82] was used to calculate MM energy. For GBSA calculations, the dielectric constant was set to 78.5 and ion concentration to 0.3 M. The surface tension was set to 0.00542 kcal/(mol·Å$^2$). The switching distance (*switchdist*) for the LJ interactions was 15 Å; the long-range electrostatic cutoff distance (*pairlistdist*) was 18 Å; and the cutoff (*cutoff*) for the nonbonded interactions was 16 Å.

## Supporting information

**S1 Fig. Flow cytometry of primary human macrophages. (a)** Successive plots and gates that were used in the flow cytometry experiments. **(b)** Source flow cytometry histogram of HLA-DR, CD86, CD206, and CD163 in primary human macrophages from 4 donors. The macrophages were gated on live CD11b population associated with the interleaved scatter plot

shown in the main Fig 1A. The underlying data for S1 Fig can be found in S1 Data.
(JPG)

**S2 Fig. Purification of the CMKLR1-G$_i$ complex with Chemerin9 and cryo-EM data processing. (a)** Size exclusion chromatography profile and SDS-PAGE analysis of the purified Chemerin9-CMKLR1-G$_i$ complex. **(b)** Representative cryo-EM micrograph (scale bar: 50 nm) and 2D class averages (scale bar: 5 nm). **(c)** cryo-EM image processing workflow for the Chemerin9-CMKLR1-G$_i$ complex. **(d)** Gold-standard FSC curve showing an overall resolution is 2.94 Å at FSC = 0.143. **(e)** Angular distribution of the particles used in the final reconstruction. **(f)** Density map according to local resolution estimation. **(g)** cryo-EM density maps and models of the 7 transmembrane helices (TM1-7), Helix 8 (H8), α5 helix of Gα$_i$, and the ligand of Chemerin9-bound CMKLR1-G$_i$ complex are shown. The EM density is shown at 0.148 threshold. CMKLR1, chemokine-like receptor 1; cryo-EM, cryo-electron microscopy; EM, electron microscopy; FSC, Fourier shell correlation.
(TIF)

**S3 Fig. Lipid molecules surrounding the 7-TMs of CMKLR1. (a)** cryo-EM density of 1 cholesterol molecule (upper, level = 0.1) and 2 palmitic acid molecules (lower, level = 1.5). **(b)** Superimposition of CMKLR1, C5aR with avacopan (PDB ID 6C1R), and GPR40 with AP8 (5TZY). The palmitic acid molecules are shown as grey sticks. Avacopan is a NAM of C5aR shown as purple sticks. AP8 is a PAM of GPR40 shown as light blue sticks. CMKLR1, chemokine-like receptor 1; cryo-EM, cryo-electron microscopy; NAM, negative allosteric modulator; PAM, positive allosteric modulator; 7-TMs, 7 transmembrane helices.
(TIF)

**S4 Fig. Mutagenesis studies. (a)** Localization of CMKLR1 variants in HEK293 cells measured by eYFP autofluorescence. Nuclei were labeled with Hoechst 33342. Scale bar: 10 μm. wt CMKLR1 is expressed only in the cell membrane as well as N191[4.77], S295[7.32], and L298[7.35]. Y471.39 is mainly expressed in the membrane but partially within the cell, whereby all 3 W273[6.48] variants are located exclusively in the cytosol. **(b)** G protein signaling of wt CMKLR1 and W273[6.48] variants stimulated by C9. The receptor variants showed no Ca$^{2+}$ release probably because of the receptor localization. **(c)** G protein signaling of wt CMKLR1 simulated by C9, [A9]-C9, and C9-NH$_2$. The amidated C-terminus has an increased influence compared to the Ala change at position 9 in C9. **(d)** Positions of Y47[1.39], N191[4.77], S295[7.32], and L298[7.35] of CMKLR1 and **(e)** signaling of their Ala variants stimulated by C9. All signaling assays were performed by measuring Ca$^{2+}$ influx. The assays were executed at least 3 times in triplicates. Results are represented as means ± SEM. The underlying data for S4A, S4B and S4E Fig can be found in S5 Data. CMKLR1, chemokine-like receptor 1; C9, chemerin9; eYFP, enhanced yellow fluorescent protein; wt, wild-type.
(TIF)

**S5 Fig. Localization of CMKLR1 variants in HEK293 cells.** HEK293 cells were transiently transfected with 1,000 ng of receptor variant fused with eYFP with Lipofectamine 2000. Receptor expression was determined by eYFP autofluorescence. Cell nuclei were stained with Hoechst33324 (blue). Scale bar 10 μm. Most receptor variants are expressed in the membrane, but Y276[6.51]A, H95[2.60]A, and H95[2.60]L are located within the cell, whereby the F88[2.53]A variant is only partially expressed in the membrane.
(TIF)

**S6 Fig. MD simulations of the CMKLR1-Gi complex. (a)** RMSD of the receptor alone and the CMKLR1-G$_i$ complex in the simulations of the CMKLR1-G$_i$ complex with (left) and

without (right) chemerin9. CMKLR1 deviated from that in the cryo-EM structure within $2.0 \pm 0.5$ Å, and the entire protein complex (CMKLR1 + chemerin9 + G protein) deviated from that in the cryo-EM structure within $3.5 \pm 1.0$ Å. In the absence of chemerin9, CMKLR1 showed more fluctuations. The underlying data for S6A Fig can be found in S6 Data. **(b)** Interfacial salt bridges intermittently formed between CMKLR1 and the $G_i$ complex during the simulations. CMKLR1, chemokine-like receptor 1; cryo-EM, cryo-electron microscopy; MD, molecular dynamics; RMSD, root-mean-square deviation.
(TIF)

**S7 Fig. Superimposition of our structure (slate) and the published structure (PDB ID 7YKD, grey) of the chemerin9-CMKLR1-$G_i$ complex. (a** and **b)** The comparison of the overall structure and the chemerin9 binding pocket, respectively. Chemerin9 in our structure and in the published structure is colored orange and yellow, respectively. The structural alignment is based on the receptor.
(TIF)

**S1 Data. The raw data for Figs 1A and S1.**
(XLSX)

**S2 Data. The raw data for Fig 4A and 4B.**
(XLSX)

**S3 Data. The raw data for Fig 5.**
(XLSX)

**S4 Data. The raw data for Fig 6A and 6B.**
(XLSX)

**S5 Data. The raw data for S4B, S4C and S4E Fig.**
(XLSX)

**S6 Data. The raw data for S6A Fig.**
(XLSX)

**S1 Raw image. The raw SDS-PAGE data for the SDS-PAGE image in S2A Fig showing the concentrated sample of the complex.**
(PDF)

## Acknowledgments

We thank Dr. Sudha Chakrapani and Dr. Kunpeng Li for oversight of the cryo-EM core facility at the Case Western Reserve University. We thank Dr. James Conway with his funding support S10 OD025009 (Krios) and S10 OD019995 (Falcon 2/3 camera) from the National Institutes of Health (NIH) in the USA for oversight of the cryo-EM facility at the University of Pittsburgh. We thank Ronny Müller, Kristin Löbner, Janet Schwesinger, and Christina Dammann at Leipzig University for excellent technical assistance.

## Author Contributions

**Conceptualization:** Mingye Feng, Annette G. Beck-Sickinger, Cheng Zhang.

**Formal analysis:** Mary Hongying Cheng.

**Funding acquisition:** Mingye Feng, Ivet Bahar, Annette G. Beck-Sickinger, Cheng Zhang.

**Investigation:** Xuan Zhang, Tina Weiß, Mary Hongying Cheng, Siqi Chen, Carla Katharina Ambrosius, Anne Sophie Czerniak, Kunpeng Li, Mingye Feng, Ivet Bahar, Annette G. Beck-Sickinger, Cheng Zhang.

**Supervision:** Mingye Feng, Ivet Bahar, Annette G. Beck-Sickinger, Cheng Zhang.

**Writing – original draft:** Mingye Feng, Ivet Bahar, Annette G. Beck-Sickinger, Cheng Zhang.

**Writing – review & editing:** Xuan Zhang, Tina Weiß, Mary Hongying Cheng, Siqi Chen, Mingye Feng, Ivet Bahar, Annette G. Beck-Sickinger, Cheng Zhang.

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
