## [Editor Report · Decision Letter 0]

7 Jun 2023

Dear Dr Zhang, 

Thank you for submitting your manuscript entitled "Structural basis of CMKLR1 signaling induced by chemerin9" for consideration as a Research Article by PLOS Biology. Please accept my sincere apologies for the long delay in getting back to you as we consulted with an academic editor about your submission. 

Your manuscript has now been evaluated by the PLOS Biology editorial staff, as well as by an academic editor with relevant expertise, and I am writing to let you know that we would like to send your submission out for external peer review under our scooping protection policy. 

Once your full submission is complete, your paper will undergo a series of checks in preparation for peer review. After your manuscript has passed the checks it will be sent out for review. To provide the metadata for your submission, please Login to Editorial Manager (https://www.editorialmanager.com/pbiology) within two working days, i.e. by Jun 09 2023 11:59PM.

Kind regards,

Richard

Richard Hodge, PhD

rhodge@plos.org

PLOS

---

## [Decision Letter · Decision Letter 1]

8 Aug 2023

Dear Dr Zhang,

Thank you for your patience while your manuscript "Structural basis of CMKLR1 signaling induced by chemerin9" was peer-reviewed at PLOS Biology. Please accept my sincere apologies for the delays that you have experienced during the peer review process. Your manuscript has now been evaluated by the PLOS Biology editors, an Academic Editor with relevant expertise, and by three independent reviewers. 

In light of the reviews, which you will find at the end of this email, we would like to invite you to revise the work to thoroughly address the reviewers' reports.

As you will see below, the reviewers are generally positive about the manuscript and note that the study is well conducted. However, Reviewer #1 raises concerns with the modelling and quality of the density maps near the extracellular side of the receptor and Reviewer #2 notes that the rationale for the study should be more clearly framed in the Introduction and Discussion sections of the manuscript. 

Given the extent of revision needed, we cannot make a decision about publication until we have seen the revised manuscript and your response to the reviewers' comments to carefully document the map quality. Your revised manuscript is likely to be sent for further evaluation by all or a subset of the reviewers.

**IMPORTANT - SUBMITTING YOUR REVISION**

*Re-submission Checklist*

*Published Peer Review*

*PLOS Data Policy*

*Blot and Gel Data Policy*

Sincerely,

Richard

Richard Hodge, PhD

rhodge@plos.org

REVIEWS:

Reviewer #1: In the article "Structural basis of CMKLR1 signaling induced by chemerin9" Zhang et al. describes the human CMKLR1-Gi complex structure bound to chemerin9, MD simulations and functional assays together to investigate the activation mechanism. Upon receiving the manuscript, I was noted that a competing paper was already published in PNAS in March 2023. Below list some of my concerns over the model and interpretations.

Major concern

1. The overall model of CMKLR1 complex is well fit to density map, but near extracellular side there is a large amount of residues missing their density. It is particularly worrying that backbone density is also missing in some areas. This could significantly mislead readers or end users of the model. Some side chains should be trimmed to avoid confusions.

2. It has been reiterated in the manuscript that the pi-pi interaction contributes to the binding of chemerin9 to the receptor. It appears to me that the authors seem to be abusing this term. For example, they stated "This residue and Phe150 of chemerin9 form π-π interactions with F2947.31 (Ballesteros-Weinstein numbering) and F190ECL2 of CMKLR1, respectively". However, Phe150 is at least 12 Å away from F294. How is any interaction possible? In addition, there is merely no density to support the side chain of Phe150. If it was indeed a good pi-pi interaction between Phe150 and F190, why was it too difficult to see its density? See the screenshot below.

Another example, "At the extracellular region of CMKLR1, the first N-terminal residue of chemerin9, Tyr149, forms a π-πinteraction with F2947.31 of CMKLR1". Although there might be a pi-pi interaction between two stated residues, it should be noted two residues at the so-called offset position are 6 Å apart. The authors need to double-check if such pi-pi interaction can be acclaimed. See the screenshot below.

Also, "Phe156 of chemerin9 plays a crucial role in the interactions with the receptor, forming hydrophobic and π-π interactions with CMKLR1 residues F882.53, H952.60, M1233.36, Y2766.51, and I3067.43…". Please check if the orientation of Phe156 relative to F88 and Y276 aligns with the definition of pi-pi interaction. 

3. The use of AlphaFold structure to represent the inactive state of CMKLR1 is unsound. First, the AlphaFold structure itself does not acclaim to be at the inactive state. Second, based on homology modeling, the source of initial models is unknown to users. Given these considerations, it is usually not used as an evidence to make comparisons between the active and inactive GPCR structures. Especially, discussing residue movements in so much details is not necessary. 

Minor concerns

1. When discussing the simulation results, the authors seem to be quite surprised that chemerin9 as a potent agonist is able to stabilize the active state of CMKLR1 in the absence of Gi. The reasons for this uncommon view are neither explained in the manuscript nor found in the reference. 

2. The authors also stated "This observation suggests that the active CMKLR1 conformation favors a close contact between the peptide agonist and the transmission switch motif." However, the two residues (Figure 6), Phe156 and W275, are at least 5.5 Å apart even when Gi is bound, which is not a close distance. Some could even say they do not contact one another. 

3. The FSC cutoff for model-to-map is 0.5 instead of 0.143.

4. Re-examining thoroughly how well the model is fit to density is necessary for this study. For example, Glu283 of the receptor is obviously laying outside its density.

5. Comparisons between current study and previously published results in PNAS should be addressed in more volumes and figures. 

Reviewer #2 (Dmitry Veprintsev, signs review): The manuscript Structural basis of CMKLR1 signaling induced by chemerin9 reports the CryoEM structure of CMKLR1 aka Chem23R in complex with one of its peptide agonists. ChemR23 is implicated in inflammation and peripheral pain. In addition to its peptide agonist, it also suggested to mediate the effects of RvE1, a pro-resolving lipid. The authors showed that inducing primary human macrophages with chemerin results in a non-canonical phenotype of the macrophages, an observation consistent with its both pro-inflammation and pro-resolving roles reported in the literature. The primary focus of the manuscript is structural analysis of the CMKLR1/Gi complex. The authors obtained a high-resolution structure, performed mutagenesis of the key residues interacting with the ligand to confirm and delaminate their role in the ligand binding or in some case also being important for activation. They also performed limited structure-activity analysis of the chemerin9 peptide, and further investigated the roles of charged interactions by MD.

Overall, this is an excellent structural biology study, providing the first experimental structure of the important pharmacological target, and a very detailed description of the ligand-binding site. Congratulations to the authors as obtaining a structure of GPCR/G protein complex is always challenging. It will be very useful for structure-based drug design of this important pharmacological target.

Addressing a few minor points would really improve the readability of the manuscript and explain its importance to a non-expert.

1. The problem statement - what exactly the authors were aiming to achieve needs to be clearly formulated. There are several problems mentioned in the introduction 1) The apparent contradictory functional roles of CMKLR1 … (2) However, it is not clear whether RvE1 can

directly bind to and activate CMKLR1 to induce Gi signaling and (3) However, the lack of synthetic ligands of CMKLR1. Which one of the problems? The solution is the structure but what was the problem?

2. There are 67 ligands reported for this receptor in gpcrdb.org as of today, although this may not have been the case when the project was initiated. Some of the argumentation may need to be revised to reflect this. Depending on the problem statement the authors would choose, this may provide an opportunity to discuss drug design in structural context.

3. A potential binding site for RvE1 suggested on Fig 3b. Again, depending on the problem statement this could be given a more prominent position in the manuscript.

4. Discussion: it is very concise but reads more like an introduction. It really needs to discuss how the new results addressed the problem statement stated in the beginning of the manuscript. Once the problem statement is clarified, it would be easier to focus the discussion. The discussion would also benefit from a short 3-4 lines conclusion summarising the key fundings.

5. Technical comment: The details of staining of the receptor in microscopy (Fig. S5) is not described anywhere. One can guess it is likely to be an anti-FLAG fluorescent Ab staining but this needs to be included in the methods.

Reviewer #3: The manuscript by Zhang et al describes the cryo-EM structure of CMKLR with chimerin9 and a Gi heterotrimeric G protein. This is second structure (and paper) describing this complex, nevertheless it is a very well performed study that adds to the understanding of how chimerin9 binds to and activates CMKLR. 

The foundation for this study is the cryo-EM map of the complex that was determined to 2.9Å resolution. This map has excellent density for all parts of the ligand and coupled with well-performed modelling creates a solid foundation for the extensive mutagenesis and ligand structure-function studies authors undertook. With the exception of some minor points listed below, I fully recommend this manuscript for a publication in PLOS Biology.

Minor comments (these are not essential for understanding the manuscript but would improve it quality ):

1. Cholesterol modelling in the map is a bit too ambitious- this density is too noisy/low contoured/uncertain to even assign a particular lipid 

2. Confusing labels on Fig. S6 as some panels have G proteins and some do not.

3. Fig. 6b- A bit weird that the cryo-EM structure did not capture the captured the closest distance of W273 and chimerin9 - a discussion sentence on that in the text would be very helpful.

4. Could the explanation for "the CMKLR1-Gi complex, the Gi heterotrimer remained stably bound to the receptor, regardless of the presence of chemerin9 (Fig. S6a)." be that the simulation is too short? Authors only run the 100ns simulation, while it is my understanding that to assess G protein binding at G proteins generally longer time frames are required (1uM). Perhaps and alternative explanation can also be added/considered.

---

## [Decision Letter · Decision Letter 2]

17 Oct 2023

Dear Dr Zhang,

Thank you for your patience while we considered your revised manuscript "Structural basis of CMKLR1 signaling induced by chemerin9" for publication as a Research Article at PLOS Biology. Please accept my apologies for the delay in getting back to you during this round of the peer review process. This revised version of your manuscript has been evaluated by the PLOS Biology editors, the Academic Editor and the original reviewers.

Based on the reviews, I am pleased to say that we are likely to accept this manuscript for publication, provided you satisfactorily address the remaining points raised by the reviewers. Please also make sure to address the following data and other policy-related requests that I have provided below (A-G):

(A) We would like to suggest the following modification to the title: 

“"Structural basis of G-protein coupled receptor CMKLR1 activation and signaling induced by a chemerin-derived agonist”

(B) You may be aware of the PLOS Data Policy, which requires that all data be made available without restriction: http://journals.plos.org/plosbiology/s/data-availability. For more information, please also see this editorial: http://dx.doi.org/10.1371/journal.pbio.1001797

-Supplementary files (e.g., excel). Please ensure that all data files are uploaded as 'Supporting Information' and are invariably referred to (in the manuscript, figure legends, and the Description field when uploading your files) using the following format verbatim: S1 Data, S2 Data, etc. Multiple panels of a single or even several figures can be included as multiple sheets in one excel file that is saved using exactly the following convention: S1_Data.xlsx (using an underscore).

-Deposition in a publicly available repository. Please also provide the accession code or a reviewer link so that we may view your data before publication. 

Figure 1A, 4A-B, Figure 5, 6A-B, S4B-C, S4E, S6A 

(C) Thank you for depositing the structural data in the PDB (8SG1) and EMDB (EMD-40450) databases. However, I note that the data is currently on hold. We ask that you please make this data publicly available before publication.

(D) Please also ensure that each of the relevant figure legends in your manuscript include information on *WHERE THE UNDERLYING DATA CAN BE FOUND*, and ensure your supplemental data file/s has a legend.

(E) For figures containing Flow Cytometry data (Figure 1A, S1), please provide the FCS files and a picture showing the successive plots and gates that were applied to the FCS files to generate the figure. We ask that you please deposit this data in the FlowRepository (https://flowrepository.org/) and provide the accession number/URL of the deposition in the Data Availability Statement in the online submission form.

(F) Please ensure that your Data Statement in the submission system accurately describes where your data can be found and is in final format, as it will be published as written there. 

(G) Please also provide a blurb which (if accepted) will be included in our weekly and monthly Electronic Table of Contents, sent out to readers of PLOS Biology, and may be used to promote your article in social media. The blurb should be about 30-40 words long and is subject to editorial changes. It should, without exaggeration, entice people to read your manuscript. It should not be redundant with the title and should not contain acronyms or abbreviations. For examples, view our author guidelines: https://journals.plos.org/plosbiology/s/revising-your-manuscript#loc-blurb

We expect to receive your revised manuscript within two weeks. 

*Published Peer Review History*

*Press*

Kind regards,

Richard

Richard Hodge, PhD

rhodge@plos.org

Reviewer remarks:

Reviewer #2 (Dmitry Veprintsev, signs review): The revised version of the paper is more focused. It reads much better, and the authors addressed most of my comments. One minor point that could still be improved (but it does not really affect the conclusions of the paper) is that there were a few small molecule antagonists reported in the literature (and GPCRDB database under the alternative receptor name CML1). These molecules may need to be mentioned for factual correctness in the introduction and corresponding section of the discussion, with the corresponding adjustment of the wording. 

https://pubmed.ncbi.nlm.nih.gov/32773087/

https://pubmed.ncbi.nlm.nih.gov/31521459/

Reviewer #3: All of my comments have been addressed.

---

## [Editor Report · Decision Letter 3]

21 Oct 2023

Dear Cheng,

On behalf of my colleagues and the Academic Editor, Raimund Dutzler, I am pleased to say that we can accept your manuscript for publication, provided you address any remaining formatting and reporting issues. These will be detailed in an email you should receive within 2-3 business days from our colleagues in the journal operations team; no action is required from you until then. Please note that we will not be able to formally accept your manuscript and schedule it for publication until you have completed any requested changes.

PRESS

Best wishes, 

Richard

Richard Hodge, PhD

rhodge@plos.org

PLOS
